# A Study on Emotions to Improve the Quality of Life of South Korean Senior Patients Residing in Convalescent Hospitals

**DOI:** 10.3390/ijerph192114480

**Published:** 2022-11-04

**Authors:** Aeju Kim, Yucheon Kim, Jongtae Rhee, Songyi Lee, Youngil Jeong, Jeongeun Lee, Youngeun Yoo, Haechan Kim, Hyeonji So, Junhyeong Park

**Affiliations:** 1Department of English Language and Literature, Dongguk University-Seoul, 30, Pildong-ro 1 gil, Jung-gu, Seoul 04620, Korea; 2Department of Counseling and Coaching, Dongguk University-Seoul, 30, Pildong-ro 1 gil, Jung-gu, Seoul 04620, Korea; 3Department of Industrial and Systems Engineering, Dongguk University-Seoul, 30, Pildong-ro 1 gil, Jung-gu, Seoul 04620, Korea; 4Dharma College, Dongguk University-Seoul, 30, Pildong-ro 1 gil, Jung-gu, Seoul 04620, Korea; 5Department of Agricultural, Wonkwang University-Iksan, 460, Iksan-daero, Iksan 54538, Korea; 6SNA-DDI, 97, Uisadandg-daero, Yeongdeungpo-gu, Seoul 07327, Korea; 7Interdisciplinary Program in Artificial Intelligence, Seoul National University, 1, Gwanak-ro, Gwanak-gu, Seoul 08826, Korea

**Keywords:** senior patient, emotion, quality of life, convalescent hospital, South Korea

## Abstract

This study examined the occurrence of emotion types and the contents and meanings of individual emotion types to improve the quality of life of South Korean senior patients in convalescent hospitals. This research is a sequential mixed study in which we conducted emotion frequency and content analyses with 20 elderly resident patients in a convalescent hospital. In the emotion frequency analysis, we performed emotion occurrence frequency analysis and clustering to create groups of subjects that showed similar distributions of emotions. The study results found that South Korean senior patients displayed six major emotions: joy, sorrow, anger, surprise, fear, and tranquility, including mixed emotional states. In the emotion content analysis, we used NVivo to categorize and analyze the interview contents based on emotion types. The study results show the characteristics of emotions according to patients’ treatment and recovery, life within narrow boundaries, relationships with new people and family, and the appearances of themselves that they could not easily but must accept. In addition, these characteristics appeared in health, environment, relationships, and psychological structures. Ultimately, the study results suggest that improving the quality of life of South Korean senior patients requires understanding of their emotions and examining diverse emotions in multiple dimensions.

## 1. Introduction

In the case of seniors, the expression of emotions is less likely to appear directly compared to other age groups [1]. Difficulties in expressing feelings lead to various psychological and physical maladjustments, which can create pathological problems and thus eventually become a factor in deteriorating the quality of life of seniors.

Investigating emotions that reflect the quality of life plays a vital role in understanding and intervening in human behaviors [2,3]. This importance is due to the fact that the most fundamental stage of emotional functions is “the form of adaptation that appears in the information processing process, which is preparation that makes people adapt themselves to environments to take actions that promote well-being” [4]. Zhang et al. [5] indicated the relationship between emotions and quality of life and suggested teaching emotion regulation strategies to seniors to improve their quality of life. In particular, to improve the quality of life of individuals in vulnerable situations, researchers should examine the characteristics of emotions. For example, Yun et al. [2] and Greenberg et al. [4] showed that the group without prior hospitalization and the group without any particular disease had a higher quality of life related to emotions than those with a history of hospitalization or illness. In addition, senior patients living in a convalescent hospital can feel emotions differently than seniors not living in convalescent hospitals. Therefore, their emotional patterns, which are different from those of ordinary seniors, are important indicators of emotion control for well-being and disease treatment through the emotion characteristics mass-produced by their diseases. Lim and Jang [6] also stated that psychological well-being, which is essential for the quality of life, is closely related to emotional control.

Convalescent hospitals are therapeutic institutions that accept geriatric and chronically ill patients and those recovering from surgery or injury [7]; seniors living in these hospitals live with various health issues such as diabetes, gastritis, depression, asthma, heart diseases, chest pain, and eye and ear problems [8]. In general, elderly patients in sanatoriums feel very lonely, and their loneliness adversely affects their health [9]. Therefore, seniors’ quality of life varies according to their emotions. In addition, social environments also affect their quality of life. For example, seniors in Germany showed increased emotional loneliness during the COVID-19 pandemic [10]. In the case of South Korea, researchers found that health-related changes in life occurred due to the pandemic [11]. However, senior patients’ perceptions of the quality of life at these senior-related institutions are not necessarily negative. According to Kang and Kim’s [12] study, elderly patients hospitalized in a convalescent hospital felt relieved because their confinement resolved their daily life problems such as obtaining clothes, food, shelter, and dealing with medical issues.

Ekman [13] defined basic emotions as joy, sorrow, fear, anger, hatred, and surprise. In addition, Kim and Min [14] studied seniors’ experiences with six emotions: anger, sorrow, disappointment, anxiety, tranquility, and joy/pleasure, based on the positive and negative affect schedule (PANAS). Therefore, the studies identified eight basic types of emotions: joy, surprise, anger, sorrow, hatred, fear, contempt, and tranquility. The different emotions vary with individuals’ perceptions or situations. Because emotions organize behavior and cognition in a certain way, classifying basic emotions and understanding their characteristics is very important in understanding the mechanisms of major human behaviors [4].

All human beings appear in various figures due to diverse socialization factors [15]. For example, a study on the quality of life of North Korean refugees reported that the events or situations they experienced affected their depression or anxiety [16]. That is, emotions are closely related to a person’s life experiences and situations. Therefore, understanding the conditions that evoke various emotions that affect the quality of life is essential. A study by Kim and Lee [17] indicated that the dimensions of difficulties experienced by the elderly in South Korea could fall into the following areas: health, social/environmental, emotional/cognitive, and economic problems. In addition, a study to validate a Korean version of the quality of life scale for the elderly indicated that factors related to the quality of their lives included controllability, autonomy, a feeling of happiness, and self-realization [18]. Moreover, Kwak [19] found that elements that significantly affect the elderly’s quality of life are health, money, work, children, spouse, and family life, in order of precedence. In addition, exercise, physical activity, and health play essential roles in maintaining seniors’ quality of life [15,20,21].

Studies related to seniors’ emotions conducted in South Korea thus far include seniors’ difficulties with [17] expression of emotions and the ambivalence of that expression in old age [1]. In addition, studies covered the effect of the elderly’s control of emotions on the quality of their lives [2], Koreans’ structure of emotion, and its measurement [3]. However, these studies are limited to the emotions of healty seniors in general. Therefore, empirical studies on the emotions and the quality of life of senior patients are insufficient.

According to a study by Jeong and Kim [22], the anxiety level of the chronically ill elderly individuals is not higher than that of senior patients admitted to the geriatric hospital. Elderly care hospitals are in charge of providing treatment, rehabilitation, and care services simultaneously to elderly patients with chronic diseases who can hardly be cared for at home and supporting them. In many cases, senior patients in a convalescent hospital are there regardless of their will. In the process, their contact with family members decreases, leading to psychosocial shrinkage and break-off of attachment [23].

When looking at study findings indicating that differences between individuals with emotion regulation disorders or social anxiety and the individuals’ circumstances affecting their quality of life [24], it is essential to look at the stories of the individuals. Therefore, in this study, we examined the emotions frequently appearing in stories and the content provoking the feelings. That is, examining what and how often senior patients feel specific emotions or what emotions they feel in what situations can provide crucial data for improving the quality of their lives. Therefore, in the present research questions, we examined the frequency of emotion types of senior patients in South Korea and explored the contents and meanings of emotion types to suggest ways to improve the quality of life of elderly patients. Our research questions are:

RQ1.What is the frequency of emotion types of senior patients residing in convalescent hospitals?RQ2.What are the contents according to the emotions of senior patients residing in convalescent hospitals?

## 2. Methods

### 2.1. Study Design

We used the sequential mixed method among mixed research methods in this study. Figure 1 [25] depicts the process of this study. After quantitative data collection and analysis, we examined the frequencies and contents of emotion types of senior patients through qualitative analysis [26].

### 2.2. Study Subjects

The senior patients in this study included those aged at least 65 years in a hospital or convalescent hospital due to illness. We used convenience sampling to select this study’s participants. In addition, we interviewed the participants face-to-face while observing the quarantine rules for COVID-19. This study received IRB approval number: DUIRB—2022108-02.

As Table 1 shows, the participants in this study were 20 senior patients, 65 years or older, in a convalescent hospital. The study participants included 9 males and 11 females in their sixties (*n* = 3), seventies (*n* = 8), and eighties (*n* = 9). From the initial recruitment, we excluded those with serious diseases that could hardly recover and those who were in a mental or emotional state in which they could not clearly express their intention despite having indicated their intention to participate in this study.

### 2.3. Data Collection and Preprocessing

This study examined the occurrence frequency and characteristics of the basic emotions of anger, joy, sorrow, surprise, fear, tranquility, hatred, and contempt to understand the psychological and emotional states of elderly patients. According to Magai et al. [27], emotion-related studies show different results because they rely on using a mixture of general frequencies and intensities or measuring the retrospective experience rate through an online evaluation system. Therefore, in this study, we identified the frequencies by analyzing the contents of responses to the questions according to the types of emotions. In addition, we explored how often the contents corresponding to each type of emotion appeared among the contents. This approach more accurately identifies the frequency of each kind of emotion based on what respondents experience rather than looking at the existing frequency scale for how often respondents feel each type.

To that end, we conducted in-depth interviews with 20 study subjects using semi-structured questions. We constructed the contents of the questions to examine the situations where emotions occur and their responses. For example, to identify the frequencies of emotions and analyze the situation in which the joy emotion occurred, the concrete contents of the joy emotions, and the reactions when the joy emotions happened, we asked questions such as “What makes you feel joy?”, “How did you react when you felt joy?”, and “How did you feel at that time concretely?” As with joy, we asked the following questions to learn the circumstances in which the sadness emotions occurred, the concrete contents of the sadness emotions, and the reactions when the sad feelings occurred: “What makes you feel sad?”, “How did you react when you felt sad?”, and “How did you feel at that time concretely?” We conducted interviews about other emotion types in a similar manner. After asking semi-structured questions about each emotion and depending on the study participants’ responses, we asked additional questions such as “Give me a concrete example” or “How can you express your emotions at that time concretely?” Three researchers trained to avoid ethical problems in the interview method conducted the face-to-face interviews while observing the COVID-19 quarantine rules.

We improved the validity and reliability of the data in this study through inter-researcher reviews [26]. Regarding data collection, three researchers were briefed in advance about the study, transcribed the recorded interviews, and labeled the emotion types and intensities for individual sentences. Each researcher independently labeled the emotion embedded in each interview sentence. We analyzed the data based on the interviewees’ meaningful values and the data entered by two or more researchers with the same emotion types. For statistical processing of the labeling of emotions and emotion intensities in interview answers, we compared and adopted the labels of the three researchers according to the following criteria.

First, when all three researchers’ evaluations were the same, we obtained the average of the three researchers’ evaluated emotion intensities. Then, we labeled the interview answer with the emotion type and the average. Second, when the evaluations of only two researchers were the same, we obtained the average of the two researchers’ evaluated emotion intensities and labeled the interview answer with the emotion type and the average. Third, when the evaluations of all three researchers were different, we set the interview answer as a non-determined value (nan) and replaced the value with the emotion type and emotion intensity value of the researcher who most matched the final emotion classification among the three researchers.

### 2.4. Data Analysis Method

In this study, we analyzed the collected data from the in-depth interviews through two methods.

(1)Frequency Analysis

The frequency analysis method identifies the general characteristics of study subjects [28]. In this study, we identified the frequency of each patient’s emotions and combinations of frequently occurring emotions.

(2)Clustering of Survey Subjects

Clustering is very effective for identifying the pattern and distribution of data [29]. It is an algorithm for unsupervised learning and refers to a method of collecting similar samples into one group. Each survey subject is expressed as a vector in the emotional dimension to cluster collected data. After adding up the emotion intensity values by emotion type that appeared in individual survey subjects, we divided the resultant value by the number of interview answers to set the value of each emotion dimension. Since there are eight emotions, the emotion vectors of the survey subject are eight-dimensional. Table 2 shows the emotion vectors of the survey subjects.

Among the clustering methods, we used k-means clustering [30] in this study. The k-means algorithm groups the given data into k clusters to minimize the variance of the differences in distances between individual clusters. In this study, we checked the most suitable number of groups while changing k from 1 to 10. In this case, we used the inertia index to find the optimal number of clusters. Inertia is the sum of squared distances between the center of a cluster and the data belonging to the cluster. Through the inertia index, we can check how much of the data belonging to a cluster is covered. However, as the number of clusters increases, the value of inertia decreases. When the number of clusters has exceeded a certain boundary, the value of inertia shows only insignificant changes. This boundary is called an elbow, and the method of clustering with the number of clusters corresponding to the elbow is the elbow technique [31].

### 2.5. Qualitative Study Procedure

The participants in the qualitative study are the same as those in the quantitative research. We used the content analysis method for qualitative analysis to find the content provoking senior patients’ emotions. The researchers used the transcribed interview data to analyze the contents according to emotion types based on the qualitative data analysis procedure of Oh, Lee, and Jung [32]. The researchers repeatedly read the contents of the answers to the questions about the eight emotions, which are the recorded interview contents, to derive meanings, give codes to the types of repetitive concepts in units of meanings, and categorize the codes by comparing individual codes. Afterward, the researchers connected the categories to form the overall subject structure.

In this study, we coded using “open coding” presented by Strauss and Glaser [33], which is a stage to “dissolve, examine, compare, conceptualize, and categorize” data. In this stage, the researcher reads the data and attaches “codes” to concepts to derive the concepts intrinsic in the data. For example, in the area of “joy”, we coded the content “It would be great if my son visits as ‘family member’s visit”. After each researcher coded independently, the researchers sought a consultation process for each coding content. If the coded contents by different researchers did not coincide, we used a third-party researcher to re-code. We used Creswell’s [34] proposed member check to improve validity in qualitative research.

### 2.6. Qualitative Data Analysis

We used NVivo12, a qualitative data analysis program, in this study. NVivo helped us manage the entire study process by storing, coding, and analyzing the vast amount of collected data to enhance data reliability. In the data analysis process in NVivo12, there are upper categories above numerous subcategories, reducing the number of categories [35]. In this study, the researchers collected meanings from vast interview data and coded them in NVivo. To re-categorize the meanings, the researchers created middle categories and coded the meanings into the middle categories. Then, the researchers carried out upper coding to upper-categorize the meanings. Two researchers carried out the coding work.

The first researcher completed the primary coding, while the second reviewed and corrected the primary coding. Then the two researchers revised the codes as necessary into agreed codes through discussion and repeated this process. Finally, the coding concluded with researchers agreeing on the final codes through revisions and discussions.

By repeating the process, they created 159 basic codes from the vast interview data (see the Appendix A, Appendix B, Appendix C and Appendix D) and 52 middle categories (Table 3), which they then grouped into four categories. In this work, the researchers coded and categorized while correcting and supplementing work through content review and meaning consideration.

## 3. Results

### 3.1. Emotion Frequency Analysis

#### 3.1.1. Frequency Analysis

The analysis results of the frequency of occurrence of emotions in this study are presented in Table 4. The most frequently occurring emotions in the 20 senior patients were joy, anger, sorrow, and tranquility, while hatred and contempt occurred relatively less often.

In addition, the study participants showed single and many emotions simultaneously. Therefore, we checked the ratios of simultaneous occurrences of emotions through support to identify the relationships between these emotions. The support in the association rule is the ratio of the number of items, including X and Y, to the number of the entire data. Table 5 shows the results of arranging the sets of simultaneously occurring emotions with support of at least 0.65 in descending order. Finally, we calculated the support values by individual investigators and added the resultant values. As a result of the support analysis, we identified that the emotions of joy, anger, sorrow, tranquility, surprise, and fear frequently occurred in the sets of simultaneously occurring emotions.

#### 3.1.2. Clustering of Survey Subjects

Regarding the clustering of survey subjects (see Figure 2), inertia variation decreased when the number of clusters was four. This result means that four clusters are the most appropriate to represent the relevant data. We intuitively checked the data distribution through visualization [36]. Figure 3 illustrates the visualization of the vector space in which we projected the vectors of 20 subjects onto the 3D space using the principal component analysis when there were four clusters. Figure 3 shows the formation of clusters divided by color. 

As a result of both analyses, we identified the most appropriate classification of emotions as four—that is, clustering analysis resulted in four clusters/emotions as the most appropriate. We checked the actual data labels and found the four clusters representing joy, anger, sorrow, and tranquility. However, based on the results of the support analysis, there were a significant number of cases where the above four emotions co-occurred with fear and surprise. Therefore, six emotions—joy, anger, sorrow, tranquility, fear, and surprise—excluding hatred and contempt are the result of the classification of emotions appropriate for the expression of the survey subjects.

### 3.2. Emotion Content Analysis

#### 3.2.1. Current Status of Four Dimensions by Emotion Type

The analysis results of the content frequency by emotion in the four dimensions of health, environment, social relations, and psychology by emotion type are presented in Table 3. We constructed the dimensions’ categories for emotions as follows: health—16 categories related to treatment and recovery, environment—13 related to life within narrow boundaries, social relations—13 related to relationships with new people and family, and psychology—10 related to the sorrow that the patient had difficulty accepting.

Then, we constructed the following emotion categories: 11 categories for joy, 14 for sorrow, nine for anger, six for tranquility, five for fear, three for surprise, three for hatred, and one for contempt. The Appendix A, Appendix B, Appendix C and Appendix D illustrates the frequency of the contents of codes by emotion.

#### 3.2.2. Analysis of Contents According to Emotion Types

The details of the four categories by emotion type of senior patients are below.

##### Health: Treatment and Recovery

The elderly patients showed different emotions according to their health status. The feeling of joy appeared as their health status improved, and they participated in daily life smoothly and exercised frequently. They showed feelings of joy just when they became able to walk while relying on a cane through rehabilitation: “I like going out to exercise” (Study Participant 3).

On the other hand, they showed sorrow for being injured when they were not very old: “My condition seemed to have improved a little when about five or six years passed after the injury, but I fell again. I collapsed. Therefore, I made a lot of effort to recover while walking with a cane. I’m not very old yet. I’m only 78 years old. Therefore, I walk around in the hospital lounge every day ” (Study Participant 2).

The senior patients regarded treatment as the most crucial element, paid attention to how their treatment was carried out and showed related emotions. However, they also showed feelings of sorrow for their missing the time of treatment: “Although I don’t know, there is something called a golden time. I missed that time. I feel as such.” (Study Participant 2).

The senior patients often expressed feelings of anger about their physical pain. They showed negative emotions about their pain not being adequately treated and were upset about the hospital’s inability to solve their pain: “I wish the hospital would give me the medicine I ask for. I feel so itchy and stuffy. They don’t let me be free or scratch as I want. I felt so itchy that I put my back on the bed and move” (Study Participant 4).

The two things that senior patients especially fear are ‘falls’ and ‘dementia’. They feared the experience of the moment they fell. In addition, many seniors around them get ill after they fall. In the case of ‘dementia’, patients are well aware of the aftereffects of dementia. In particular, we found that they thought dementia would cause suffering for themselves and their family members. Study Participant 8 talked about her fear of dementia as follows: “Dementia really hurts families. I thought that I should not get dementia until I die. While seeing someone with dementia, I thought I shouldn’t do that. What I fear the most is dementia” (Study Participant 8).

##### Environment: Life in Narrow Boundaries

The senior patients enjoyed eating their favorite foods even in their life within the narrow boundaries termed hospital. They felt joy when their family members brought them food they liked at ordinary times and ate it: “I feel good when my younger siblings packed and brought what I like such as bread or rice cakes because they know what I like” (Study Participant 8).

In addition, as they learned new things and searched for exciting stuff through YouTube, they had feelings of joy through daily hobbies: “I watch YouTube a lot on my smartphone when I have time, and through it, I search for things I don’t know, so I spend a lot of my time watching YouTube on my own. In particular, I watch it a lot because what people are talking about is interesting. I just spend so much time on it to the extent that my time is insufficient, and the battery runs out. I learn while realizing there are things I don’t know” (Study Participant 8).

Meanwhile, the senior patients showed feelings of sorrow about economic difficulties. They wanted to help others while they lived if they had enough money, but seeing themselves sick and without money, they felt sadness. In addition, concerning hospital expenses, they felt sorrow thinking about being deprived of their money. They felt sorrow and contempt at the thought that since they had no car and money, they could not go anywhere, even if they wanted to go somewhere: “Really, neither my children nor I have any money, but I think the money we had was used by the hospital as they like. It happened in other hospitals than here, too. It was really hard because of the people who took all the hospital expenses” (Study Participant 12).

The uncomfortable space of the hospital per se made senior patients angry. Study Participant 17 said the hospital bed was uncomfortable because it was not his bed. Study Participant 18 hated the smell of the hospital and the smell of the people around him. The senior patients lived with negative emotions within the narrow boundaries. They also showed feelings of fear and hatred when people around them were shouting. In addition, they expressed negative emotions about the behavior of patients living together in the hospital: “Crazy persons scream so badly, and they just do it. Then I am scared. I just stay still” (Study Participant 6) and “A person next to me, there was an old woman who was strange. Whenever I said something, she shouted; whenever I said something, she said it was noisy. While she was in the room, we fought for a few days, and I went to another room” (Study Participant 20).

##### Social Relations: Relationships with New People and Family

The senior patients living in the hospital showed positive and negative emotions in new relationships with hospital staff. For instance, they experienced joy and trust when they felt that the hospital staff’s attitudes toward treating senior patients were kind and reliable: “After I was hospitalized, when there was the former department manager, he was very kind, and the nurses, the director, and the caregivers all took good care of me. So, no matter what I should do, I worked hard unconditionally. I felt good, and I laughed” (Study Participant 9).

When the senior patients felt positive emotions in their relationship with hospital officials, they thought of them as their family members (Study Participant 4, Study Participant 9). When the hospital staff treated them well and nodded to them in greeting, the patients felt more joy. Among hospital staff, the nurses and caregivers often meet with senior patients in person. The patients feel joy and gratitude when these hospital staffs treat them well: “Well, the caregiver is the best” (Study Participant 13), and “The caregiver brings food to the restaurant, I just eat it, and then he cleans up everything. If I can’t go to the bathroom, he helps me now, the caregiver. I appreciate him” (Study Participant 7).

However, we found that they distrust hospital staff and express dissatisfaction when the hospital staff does not satisfy their needs: “For a year, I was shaking and swinging my legs because I was sick, and they tied me up here. When I think of that, I tremble” (Study Participant 1). Study Participant 14 said, “I got angry many times I get angry because of the caregiver. Some caregivers don’t even pretend to hear when I ask for water; they just move around.” In addition, Study Participant 9 stated, “I don’t even want to think about the caregiver who was here last time.”

Senior patients still showed a lot of content with their emotions about separated family members. They expressed joy when their family visited them and were living well; they expressed more joy especially when their grandchildren did well: “I am grateful that my children are living well without any problems. So, I’m happy. Our eldest son has two daughters, and our eldest granddaughter got a job at a hospital in Seoul during a difficult time like these days. My daughter-in-law and son are doing well. I am grateful to the extent that I cannot express it” (Study Participant 20). In addition, Study Participant 8 stated, “I am very happy when I see my grandson, and I appreciate it. I think that they came here despite that it should be very difficult for them to come here and go back as such.” Study Participant 10 said, “What makes me happy is just my grandchildren’s marriage, and when that happens, I’m happy and really grateful.”

The senior patients expressed joy about meeting their family members and grandchildren. However, they regretted and felt sorry for not being able to see their family members properly or not having done well for their family members. They were sad when they longed for or worried about their family members or got angry due to challenging family relations: “It’s been a long time since I saw my eldest son because he did not come to see me. He does not call me even once despite that he is not busy. The second son is busy and does not come often. He has become a company manager. Since I can’t go out as I like, I gave up seeing my children. When I call, they don’t answer because they’re busy” (Study Participant 10).

Some senior patients also regretted not doing well with their children: “I now feel that I was too strict to my children as their father” (Study Participant 19).

Although family members give joy, some senior patients expressed anger due to sour relations with their family members: “I got divorced because I didn’t get along with my husband, and I took my two sons without preparation, but my eldest son treats me poorly as he has a foul temper” (Study Participant 4).

##### Psychology: Accepting My Appearance, but It Is Not Easy

We found that senior patients feel joy when they accept their situation and appearance in a limited environment and have a positive view of others: “Most things are fine to be. Since everything is positively made by nature and disappears, you can just leave it as it is. Because I think positively, there is nothing particularly bad or good. I am fine today too. I live without anger. If I live with anger, that will be bad for me. Therefore, why should I be angry?” (Study Participant 18).

While accepting the situation, the senior patients also showed a mind to help and give something to those in need: “When there are persons in need, I hand out food trays. I am the oldest. Even though I am the oldest, I can do it, and I do it when I want to. Of course, I do it because I want to do it. There are persons who can’t walk well. Because there is no caregiver, they can’t walk, and that is why I do it” (Study Participants 18).

However, we found that senior patients feel lonely when they no longer feel hope, and they feel sorrowful and get angry when they feel resentment toward others: “I don’t want to live. There will be no hope if my body goes wrong further. Well, I want to die” (Study Participant 4). In addition, Study Participant 15 said, “I want to die. Even when I’m sad, I just cry alone. I think I am miserable.”

The senior patients feel that their pride is hurt; they are sad and get angry when psychological conflicts occur in their relationships with others: “There are men younger than me, and when I watch TV with them, they seem to ignore me because I am older. They didn’t talk to me openly as such, but I felt that they were talking to me as such. I don’t think that those young men do not get old! There is an order when people are born, but there is no order when people die! I think this way by myself. When patients see persons who are a little weaker than them, they despise the persons a little” (Study Participant 2).

The senior patients felt lonely at their spouse’s death and more sorrow and loneliness about living alone: “When my wife died, I thought I should have died first” (Study Participants 7). In addition, Study Participant 2 stated, “About three or four months before my husband died, in fact, when he was alive, I thought that I would be comfortable if he died soon. But still, I realized that it is very important to live as a married couple relying on each other. After my husband died, I felt so lonely and solitary, and I couldn’t stand it because I didn’t have any children” (Study Participant 2).

## 4. Discussion

This study examined the frequency of occurrence and content characteristics of the emotion types of South Korean senior patients. Study 1 found that hatred and contempt did not occur as frequently as anger, joy, sorrow, surprise, fear, and tranquility emotions for senior patients. Therefore, senior patients feel less extreme negative emotions.

In addition, the frequency of positive emotions was high even in environments where highly vulnerable seniors live, such as convalescent hospitals. According to a study by Kim and Min [14], the intensity of positive emotions increases with age, and the intensity of negative emotions decreases with age, except in the very old age group. This finding is similar to the results of this study. In addition, senior patients simultaneously feel conflicting emotions, such as joy and sorrow. This finding of the exhibition of such mixed emotions is similar to a study on emotional experiences in old age [14].

According to a study by Choi and Choi [3], emotions such as “achievement” or “envy and jealousy” also appear in South Korean culture, inconsistent with this study’s results. Furthermore, the senior patients in the present research showed their emotions in the living environment in the convalescent hospital rather than accomplishing or wishing for something, thereby indicating that general senior population’s and senior patients’ emotion characteristics differ.

In Study 2, we examined the emotional characteristics of senior patients. Based on the preceding, we provide the following insights into the issues related to the quality of life of senior patients. First, the health status of elderly patients is an important aspect that determines their emotions. Their subjective health affects their quality of life [37]. This finding is similar to a study by Menec and Chipperfield [38], indicating that the positive emotions felt by senior patients when they exercise for health or rehabilitation can improve their satisfaction with life. In addition, rather than mental illness, physical illness leads to the choice of voluntary assisted death (VAD) [39]. Therefore, above all, health-related situations affect the lives of senior patients.

First, senior patients fear “falls” and “dementia”; they fear what causes secondary physical danger in their present state. Researchers [40] have shown that a fall prevention program for the elderly with dementia and caregivers in a convalescent hospital improves physical and cognitive functions, reduces depression, and reduces psychological and behavioral symptoms. Therefore, a preventive program that can relieve their fear is necessary to enhance senior patients’ quality of life.

Second, senior patients experience changes in their emotions due to the situations and events that occur because they live in the limited environment of a convalescent hospital. The hospital life of seniors means disconnection from their everyday life. They experience changes in their physical environment from where they have lived thus far and changes in their daily life patterns and social support networks [41]. Therefore, it is necessary to pay attention to their positive emotions in simple activities such as watching YouTube. Allowing them to use the Internet in their limited environment can be a way to increase their positive emotions.

According to a study with Swiss seniors, Seifert and Schelling [42] found that most participants said that the Internet is not a waste of time and should be used. However, a study by Namazi and McClintic [43] revealed when seniors living in long-term care institutions did not use computer-based devices, they experienced limitations in their cognitive, personal, and physical environments. Therefore, it is necessary to provide education or guidance on the use of the Internet suitable for the circumstances and characteristics of senior patients to improve the quality of their daily lives.

We conducted this study in 2021, during the COVID-19 pandemic. Therefore, external factors impacted the emotions of senior patients during our investigation. For instance, in a survey of the life experiences of seniors living in India under COVID-19 lockdown conditions, the researchers found that the patients experienced anxiety, fear, and mental trauma [44]. Thus, the senior patients in our study underwent a similar situation to the Indian research. Moreover, according to Fry [45], seniors want independence and autonomous decision-making. The results of our research support such studies showing the psychological characteristics demonstrated in the unexpectedly changed environment.

Third, because they were in a convalescent hospital, the senior patients made new relationships with unfamiliar doctors, nurses, nurses’ aides, and caregivers and displayed various emotions in such relationships. In supportive social interactions, the mere presence of surrounding people who are not relatives or friends is beneficially associated with the mental health of seniors regardless of the role or function of the surrounding people [46]. This finding means that the relationship between senior patients and hospital staff is essential; forming relationships with them can improve senior patients’ quality of life.

However, the senior patients still showed deep feelings toward their family members, such as longing for the family members and joy at their visits. A study with senior patients in convalescent hospitals showed that the lower the family support, the higher the death anxiety [47]. In particular, the senior patients showed a lot of affection for their grandchildren and great-grandchildren. This affection reflects the characteristics of Korean culture.

Fourth, the senior patients showed different emotions depending on whether they accepted their situation in the hospital. For instance, Erikson [48] said that the eighth stage, corresponding to the senior period, is a period of “integration versus despair” and that if the integration of self forms in this period, it will be satisfactory. Healthy seniors can show shame about the “loss of face” [49]. Accepting their situation and openly expressing such feelings can improve their quality of life. In addition, we found that senior patients feel a lot of loneliness. According to Jansson et al. [9], we can view seniors’ loneliness in long-term care facilities (LTCF) in social, emotional, and existential aspects, and all these aspects appear in this study.

Finally, we classified the elements affecting the emotions of senior patients into health, environment, social relations, and psychology. This classification is similar to “health, social environment, economy, and emotional cognition” identified as difficulties experienced by the elderly in South Korea, developed by Kim and Lee [17], reflecting the characteristics of senior patients living in convalescent hospitals.

### 4.1. Recommendations

Based on the results of this study, we offer the following suggestions.

First, our study shows that South Korean seniors might feel negative emotions because they still perceive that in the convalescent hospital environment, they are not cared for by their family members. In Bowman and Singer’s [50] study on the perception of “advance directives” with senior Chinese patients living in Canada, the researchers mentioned cultural differences due to the influence of Confucianism, Buddhism, and Taoism. South Korea has a strong culture of filial piety, where children support their parents due to the influence of Confucian culture. Therefore, in Korean culture, parents feel disappointed, and children feel guilty when their senior parents no longer live with their children. However, it is necessary to improve the quality of life of seniors by helping them understand that “filial piety” can be applied differently from the traditional method.

This study showed the emotional characteristics of senior patients appearing in Korean culture. The results reflect that the emotions of senior patients residing in convalescent hospitals differ according to culture, thereby preparing a basis for comparison with senior patients in other cultural areas. In studies hereafter, examining the emotions of senior patients in various cultural areas is necessary by comparing them with those of senior patients in convalescent hospitals in other countries.

Second, we found that, although senior patients need to adapt to new people in a convalescent hospital, their relationships with family members are still significant. The connections can only be improved when the perceptions of both parties are known. In a study on caregivers of a senior patient with a chronic illness, the researchers found that the caregivers had difficulties in areas such as the impact of caregiving on family relationships, disruption of social relationships, disruption of personal and occupational plans, physical health-related issues, negative emotions, and dealing with the high costs of living [51]. This study did not examine the perceptions of family members who care for senior patients. However, it is also necessary to explore caregivers’ perceptions to prepare a plan to improve the quality of life of senior patients.

Third, according to a study [52] to improve patients’ survival and quality of life, the researchers identified the necessity of physical/emotional support from partners, friends, and health professionals, including emotional and social support, health policy, transportation, and job retention. In addition, a study by Manju [53] found an association between emotion regulation and quality of life. In this study, too, we saw that when senior patients were aware of their emotions and regarded that they were receiving emotional and social support from their surroundings, this state impacted their quality of life. We also found that individual and environmental characteristics affect the quality of life [16].

Therefore, an integrated approach to emotional and social support is necessary to alleviate the negative emotions felt by senior patients.

### 4.2. Limitation

The participants in this study were senior patients residing in a convalescent hospital. In selecting the participants, we used convenience sampling because of the limited accessibility to senior patients in convalescent hospitals due to COVID-19. Therefore, it is difficult to guarantee the homogeneity of characteristics, except that the subjects of this study were living in the same convalescent hospital. In addition, since we only studied the characteristics of senior patients residing in a unique environment termed convalescent hospitals, future research could focus on the emotions of senior patients living at home. Moreover, we based this study on the eight emotions regarded as basic according to the theoretical basis. Therefore, future studies could investigate more than just basic emotions. In addition, because our study included senior patients of at least 65 years old, a study that can reflect age-related characteristics is necessary for a more in-depth understanding of the emotions of elderly patients.

## 5. Conclusions

South Korean senior patients displayed six major emotions: joy, sorrow, anger, surprise, fear, and tranquility; mixed emotional states also appeared. They described the characteristics of emotions regarding to the following: their treatment and recovery, life within narrow boundaries, relationships that began again, family members, and their appearances that they could not easily but must accept. These characteristics of emotions appeared in the structures of health, environment, relationships, and psychology. Based on this research, it is necessary to seek ways to improve seniors’ quality of life according to emotions. In particular, emotion-based healing is vital for senior patients who display negative emotions such as sorrow, surprise, and fear.

## Figures and Tables

**Figure 1 ijerph-19-14480-f001:**
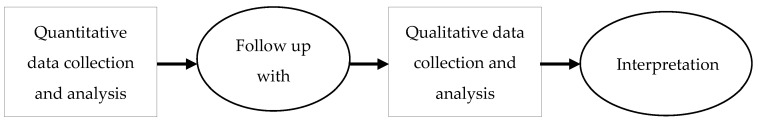
Explanatory Sequential Design.

**Figure 2 ijerph-19-14480-f002:**
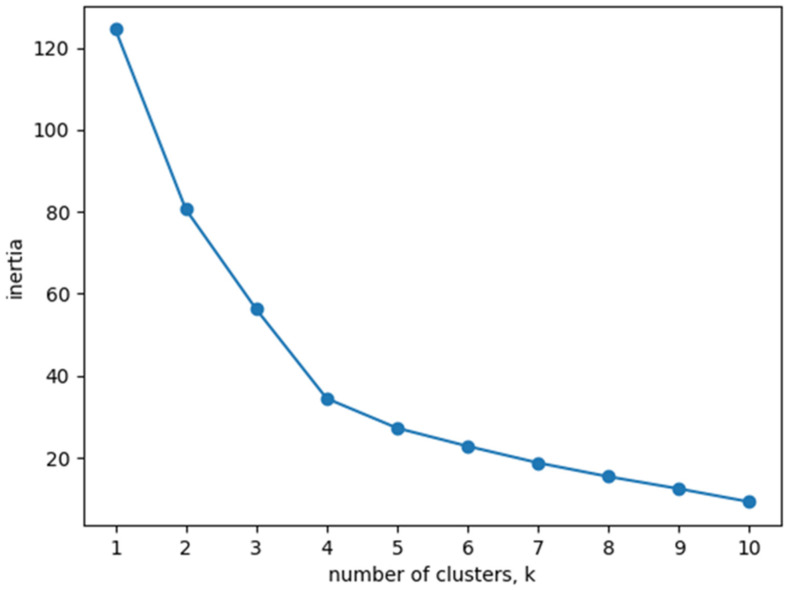
Graph of inertia values according to the number of clusters.

**Figure 3 ijerph-19-14480-f003:**
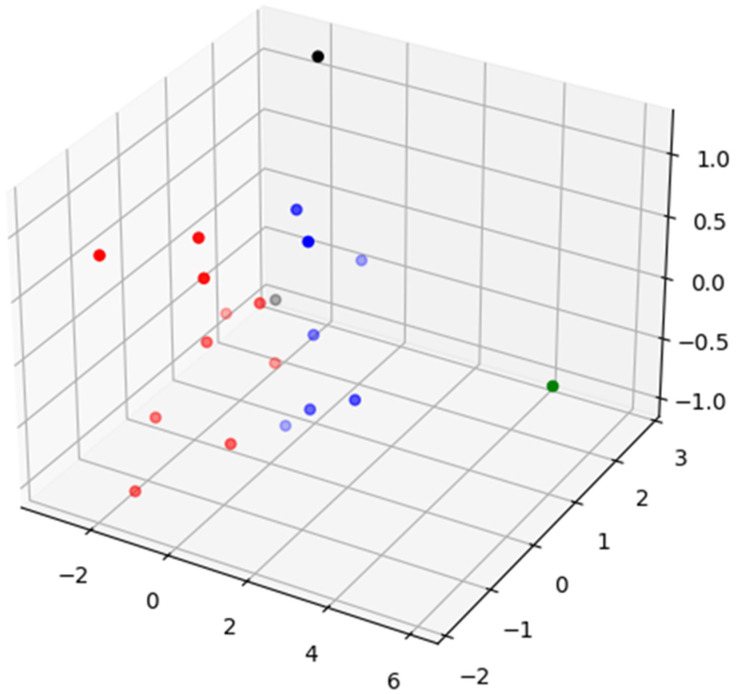
Visualization of the clusters of survey subjects after performing K-means (K = 4).

**Table 1 ijerph-19-14480-t001:** Demographic background of senior patients.

No	Gender	Age	Major Source of Income	The Persons Most Frequently Met Other Than Hospital Personnel after Hospitalization	Reason for Hospitalization (Including Disease Name)	Length of Hospitalization
1	Male	72	Farming	Children	Pain in the arm	2 years
2	Female	78	Other (recipient)	Younger sister/brother	Cerebral infarction	At least 5 years
3	Female	88	Other (unknown)	Son	Unknown	Unknown
4	Female	73	Other (compensation)	Caregiver	Unknown	Unknown
5	Male	66	Other (worker’s compensation)	Children	Cervical spine surgery	At least 5 years
6	Female	81	Family members such as children	Children, friends	Fall at home	Unknown
7	Male	78	Guard’s work	Children, friends	Not remembered	1 year
8	Female	66	Pension	Younger sister/brother	Cerebral hemorrhage	At least 3 years
9	Female	87	Other (recipient)	Grandson, granddaughter	Rehabilitation after hip surgery	Less than 2 years
10	Female	82	Family members such as children	Children	Stroke, diabetes	5 years
11	Female	86	Family members such as children	Daughter-in-law	Surgery due to a leg injury	Unknown
12	Male	71	Other (unknown)	None	Stroke	5 months
13	Male	85	Pension (severance pay)	None	Heart surgery	20 years
14	Female	80	Other (unknown)	Children	Parkinson’s disease	7 years
15	Female	77	Pension	Caregiver	Cause unknown	A few days
16	Male	73	Pension	Children	Liver transplant, hernia	1 year
17	Male	86	Pension, assets, tangerine farming	Caregiver	Cerebral infarction	1 month
18	Male	86	Family members such as children	Children	Cerebral infarction	At least 5 years
19	Male	67	Self-employment	Church friends	Spinal and limb paralysis due to car accident	At least 7 years
20	Female	76	Family members such as children	Child (eldest son)	Rehabilitation after leg fracture	At least 1 year

**Table 2 ijerph-19-14480-t002:** Emotion vectors of 20 survey subjects.

Study Participant	Emotion							
	Joy	Surprise	Anger	Sorrow	Hatred	Fear	Contempt	Tranquility
1	1.25225	0.89189	0.18919	1.61261	0.00000	0.78378	0.00000	1.25225
2	2.71429	0.00000	4.62698	0.58730	0.00000	0.00000	0.00000	0.00000
3	0.00000	0.41270	0.00000	8.25397	0.00000	0.44444	0.00000	0.00000
4	1.12121	0.02273	0.00000	4.62879	0.00000	0.00000	0.00000	0.40530
5	0.69697	0.37879	4.16667	1.31818	0.00000	0.00000	0.00000	1.30303
6	1.97619	0.00000	0.00000	0.32143	0.00000	0.00000	0.00000	2.64286
7	2.12121	0.00000	2.04545	1.08333	0.00000	0.00000	0.07955	0.69697
8	2.55758	0.32727	0.29091	2.64242	0.60606	0.33333	0.00000	0.96364
9	1.39247	0.10753	0.93190	2.25806	0.04301	0.60215	0.53405	0.67563
10	1.08835	0.43373	0.70683	3.20482	0.01205	0.01205	0.01205	0.01205
11	1.36735	2.17347	0.00000	1.62585	0.02041	0.77551	0.02041	0.02041
12	2.84459	0.01351	0.78378	0.79279	0.00000	0.01351	0.60811	0.93694
13	1.86404	0.02193	0.88158	1.25877	0.19737	0.65351	0.01316	0.38596
14	1.54000	0.36889	0.15111	3.74222	0.00000	0.08889	0.00000	0.22667
15	0.49383	0.00000	1.34568	3.60802	0.00000	0.33951	0.00000	0.19753
16	3.03226	0.00000	0.00000	0.78495	0.00000	0.17742	0.00000	0.38710
17	1.83987	0.00000	1.43791	2.26797	0.00000	0.26797	0.00000	0.14706
18	1.78947	0.00000	0.49123	3.02193	0.00000	0.00000	0.00000	0.00000
19	0.42593	0.30093	1.03241	2.46759	0.19907	0.40741	0.07407	0.43981
20	0.24883	0.14554	0.24883	3.13615	0.06103	0.09859	0.11033	0.56103

**Table 3 ijerph-19-14480-t003:** Four dimensions by emotion type of senior patients and categories by type.

No.	Division	Emotion								Category Total
		1	2	3	4	5	6	7	8	
1	Health: Treatment and recovery	Health status, daily life functions, exercise, treatment	Difficulty in daily life, lack of motivation, loss of health due to accident	Pain due to the disease, dissatisfaction with treatment	Effort for health	Accidents, treatment, symptoms appearing due to the disease	Treatment process, painful symptoms	Uncomfortable body		16
2	Environment: Life within narrow boundaries	Food intake activities, daily life, hobbies	Discomfort, financial difficulties, hospital costs	Discomfort, lack of an appropriate treatment environment, and lack of desired routines	Religion	Surrounding people, environment	Surrounding environment	Surrounding environment		13
3	Social Relations: Relationships with new people and family	Relationship with family members, relationship with hospital staff	Attitude toward family members, relationship with family members, attitude toward hospital staff, conflict with children	Relationship with family members, conflicts in the hospital	Trust in people, dependence on people	Negative impact on family members		Conflict with patients	Conflict in hospital	13
4	Psychology: Sorrow that can hardly be accepted	Positive attitude, receptive attitude	Lethargy, negative attitude toward self, worry about the situation, psychological conflict with others	Attitude toward the outside, inner attitude	Attitude toward others, inner attitude					10
Category Total		11	14	9	6	5	3	3	1	52

Emotion Legend: 1 = Joy, 2 = Sorrow, 3 = Anger, 4 = Tranquility, 5 = Fear, 6 = Surprise, 7 = Hatred, 8 = Contempt.

**Table 4 ijerph-19-14480-t004:** Frequency of occurrence of emotions by survey subject.

Study Participant	Emotion							
	Joy	Surprise	Anger	Sorrow	Hatred	Fear	Contempt	Tranquility
1	6	4	2	14	0	5	0	6
2	8	0	11	2	0	0	0	0
3	0	1	0	19	0	1	0	0
4	8	1	0	30	0	0	0	5
5	2	1	12	4	0	0	0	3
6	6	0	0	1	0	0	0	7
7	15	0	13	10	0	0	1	5
8	19	2	2	19	5	2	0	6
9	19	1	13	34	2	8	8	8
10	14	7	14	44	1	1	1	1
11	13	13	0	16	1	4	1	1
12	36	1	12	8	0	1	7	9
13	26	1	9	19	3	8	1	9
14	20	4	2	45	0	1	0	3
15	5	0	11	32	0	4	0	2
16	20	0	0	6	0	2	0	3
17	16	0	12	19	0	2	0	2
18	13	0	4	21	0	0	0	0
19	6	4	13	34	3	5	1	6
20	4	2	4	47	2	1	2	9
Total	256	42	134	424	17	45	22	85

**Table 5 ijerph-19-14480-t005:** Association rule support analysis.

Number	Support	Emotion and Emotion Pair
1	1	Sorrow
2	0.95	Joy
3	0.95	Sorrow, Joy
4	0.85	Tranquility
5	0.85	Sorrow, Tranquility
6	0.85	Tranquility, Joy
7	0.85	Sorrow, Tranquility, Joy
8	0.75	Joy, Anger
9	0.75	Sorrow, Anger
10	0.75	Anger
11	0.75	Sorrow, Joy, Anger
12	0.7	Fear
13	0.7	Sorrow, Fear
14	0.65	Sorrow, Tranquility, Joy, Anger
15	0.65	Sorrow, Fear, Joy
16	0.65	Sorrow, Tranquility, Fear
17	0.65	Sorrow, Tranquility, Anger
18	0.65	Tranquility, Fear, Joy
19	0.65	Tranquility, Anger
20	0.65	Tranquility, Joy, Anger
21	0.65	Tranquility, Fear
22	0.65	Surprise
23	0.65	Sorrow, Surprise
24	0.65	Fear, Joy
25	0.65	Sorrow, Tranquility, Fear, Joy

## Data Availability

The authors can provide data upon request.

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
