# Peer review of "A Study on Emotions to Improve the Quality of Life of South Korean Senior Patients Residing in Convalescent Hospitals"

_ijerph, 2022, doi:10.3390/ijerph192114480_

Round 1
Reviewer 1 Report
Dear authors. I have read the manuscript and consider that while it has positive aspects it also has limitations that I suggest be reviewed in depth.
Among the positive aspects, I highlight the intention to combine quantitative and qualitative analysis of the data.
In terms of limitations, I would highlight the following:
- I believe that the logic of the arguments developed in the introduction needs to be revised. I believe that the contribution that this study seeks to make to the previous literature is not adequately specified. This is mentioned but with little depth. Why is it believed that people in this type of residence will present particularities not present in other contexts? What things are expected to be consistent and what things are expected to be divergent between different inpatient and outpatient contexts? How do you reconcile the thesis that emotions are universal with the thesis that they vary according to context (both theses are formulated in the introduction)? Related to the previous question, what is expected to vary, the emotion or the triggers or stimuli of those emotions? And ultimately, how can the results derived from this research shed light on the international literature on emotions and quality of life?
- Regarding the method: a) It does not seem accurate to indicate that there are two studies. In fact, it appears to be a single study at a single point in time, where a sequential mixed data analysis was applied. b) The selection of the sample is neither clarified nor justified. c) The sample is highly heterogeneous, which makes it difficult to understand how people in this type of residence can be considered homogeneous. In other words, the heterogeneity described with regard to pathologies and years of hospitalisation makes it very questionable whether it is possible to speak in general of old people in "convalescent hospital". d) Why did interest focus only on the emotions described and not on other possible emotions? Why was this sequential mixed method based on interviews chosen instead of, for example, standardised scales? How were the questions selected and why was it decided to use these questions instead of others? It is not clear to me how from these general questions referring to concrete emotions about "What makes you happy? How did you react when joy arose? What makes you feel sad?" can infer different emotions such as "anger, sorrow, surprise, fear, tranquility, hatred, and contempt". e) It is difficult for me to understand how from textual responses of the participants the researchers determined such a variety of emotions as well as their intensity. In any case, in order to make the process transparent, it would be useful to give access to the raw material as well as to the processed material. In this way it will be possible to verify more clearly how this was done. I would also suggest providing data on inter-rater consistency or agreement. Related to this, when more than one emotion was detected, how can it be determined that this is what the older person experienced or does it refer to the divergence in how the researchers interpreted a given interview-text fragment? Why was no method used to check the validity of the coding done by the researchers? In the latter sense, serious doubts remain as to whether what is coded represents what is experienced by the interviewees.
I hope that from these suggestions the authors can revise and improve their report.
Author Response
- I believe that the logic of the arguments developed in the introduction needs to be revised.
Authors’ Response : We have made major revisions to the Introduction section as advised.
- I believe that the contribution that this study seeks to make to the previous literature is not adequately specified. This is mentioned but with little depth.
Authors’ Response : We have added these details as advised. Please refer to the Introduction and Recommendations sections.
- Why is it believed that people in this type of residence will present particularities not present in other contexts?
Authors’ Response : We have added these details as advised. Please refer to the Introduction section.
- What things are expected to be consistent and what things are expected to be divergent between different inpatient and outpatient contexts?
Authors’ Response : We have added for clarification accordingly. Please refer to the Introduction section.
- How do you reconcile the thesis that emotions are universal with the thesis that they vary according to context (both theses are formulated in the introduction)?
Authors’ Response : We have revised it for clarification. Please refer to the Introduction and Methods sections.
- Related to the previous question, what is expected to vary, the emotion or the triggers or stimuli of those emotions?
Authors’ Response : The purpose of this study was to examine the frequency of emotions according to the types of emotions and analyze the contents where these emotions appear, that is, what stimulates these emotions. We have added these details for clarification. Please refer to the Introduction and Methods sections.
- And ultimately, how can the results derived from this research shed light on the international literature on emotions and quality of life?
Authors’ Response : We have added these details as advised. Please refer to the Recommendations section.
- It does not seem accurate to indicate that there are two studies. In fact, it appears to be a single study at a single point in time, where a sequential mixed data analysis was applied.
Authors’ Response : We have revised it as advised. Please refer to the Methods section.
Thank you.
- The selection of the sample is neither clarified nor justified.
Authors’ Response : We have revised it for clarification as advised. Please refer to the Methods and Limitation sections.
- The sample is highly heterogeneous, which makes it difficult to understand how people in this type of residence can be considered homogeneous.
In other words, the heterogeneity described with regard to pathologies and years of hospitalisation makes it very questionable whether it is possible to speak in general of old people in "convalescent hospital".
Authors’ Response : Thank you for your comment. We have revised it for clarification as advised. Please refer to the Methods and Limitation sections.
- Why did interest focus only on the emotions described and not on other possible emotions?
Authors’ Response : Based on theoretical basis, we conducted this study based on the eight emotions regarded as basic emotions. Further research on emotions other than the basic emotions would be recommended. Therefore, we have added this in the Limitation section.
- Why was this sequential mixed method based on interviews chosen instead of, for example, standardised scales?
Authors’ Response : We have provided futher explanation regading this approach for clarification in the Data Collection and Preprocessing section.
- How were the questions selected and why was it decided to use these questions instead of others?
Authors’ Response : We have revised it for clarification. Please refer to the Methods section.
- It is not clear to me how from these general questions referring to concrete emotions about "What makes you happy? How did you react when joy arose? What makes you feel sad?" can infer different emotions such as "anger, sorrow, surprise, fear, tranquility, hatred, and contempt".
Authors’ Response : We have revised it for clarification and better understanding. Please refer to the Methods section.
- It is difficult for me to understand how from textual responses of the participants the researchers determined such a variety of emotions as well as their intensity.
In any case, in order to make the process transparent, it would be useful to give access to the raw material as well as to the processed material.
In this way it will be possible to verify more clearly how this was done.
I would also suggest providing data on inter-rater consistency or agreement.
Authors’ Response : We can provide recording files and transcription data upon request. Also, we have revised the Methods section as advised for clarification.
- Related to this, when more than one emotion was detected, how can it be determined that this is what the older person experienced or does it refer to the divergence in how the researchers interpreted a given interview-text fragment?
Authors’ Response : We have added further explanation for better understanding. Please refer to the Methods section.
- Why was no method used to check the validity of the coding done by the researchers?
Authors’ Response : We did explain the method we used to check the validity previously in the Data Collection and Preprocessing section. We have additionally specified the type of method for clarification. Thank you.
- In the latter sense, serious doubts remain as to whether what is coded represents what is experienced by the interviewees.
Authors’ Response : We have revised it for better understanding. Please refer to the Methods section.

Reviewer 2 Report
Introduction, line 70. Make a transitional sentence to connect the paragraphs in terms of thread.
Materials and Methods. Line 104. not described: the study design as well as the inclusion criteria.
From lines 133 to 141: what was the criteria to determine this type of qualification and information processing?
Recommendations. Line 491. Be careful with generalizations given the type of study.
Some references are not cited. For example: Line 604
Author Response
- Introduction, line 70. Make a transitional sentence to connect the paragraphs in terms of thread.
Authors’ Response : We have revised it accordingly. Please refer to the Introduction section.
- Materials and Methods. Line 104. not described: the study design as well as the inclusion criteria.
Authors’ Response : We have added these details as advised. Please refer to the Methods section.
- From lines 133 to 141: what was the criteria to determine this type of qualification and information processing?
Authors’ Response : We have additionally specified the type of method for clarification. Thank you. Please refer to the Methods section.
- Recommendations. Line 491. Be careful with generalizations given the type of study.
Authors’ Response : We have revised it as advised. Thank you.
- Some references are not cited. For example: Line 604
Authors’ Response : We have revised it accordingly.

Round 2
Reviewer 1 Report
Dear authors, I have reviewed the responses and additions made to my comments and found them to be accurate and pertinent. As a final comment:
- It would be important to clarify which instruments were used in the "Quantitative data collection and analysis" phase. From what is written in the manuscript, it is not clear whether the initial frequency analysis and cluster analysis are based on the same data generated from the in-depth interviews or on data generated from other instruments. In addition to clarifying this point, it would be good to include the data collection instruments used as supplementary material.
I wish you all the best in your work.
Author Response
Thank you for your feedback.
There was only one set of data that we collected and used for the analysis methods.
So, we have clarified this in our manuscript as advised.
We have the informed consent form and the interview questions translated in English, as well as the transcription file of the audio recordings of the interviews in Korean. We have submitted these supplementary files as advised.
